# Expansive data, extensive model: Investigating discussion topics around LLM through unsupervised machine learning in academic papers and news

Hae Sun Jung[1], Haein Lee[1,2], Young Seok Woo[1], Seo Yeon Baek[3], Jang Hyun Kim[1,2,3]*

1 Department of Applied Artificial Intelligence, Sungkyunkwan University, Seoul, Korea, 2 Department of Human-Artificial Intelligence Interaction, Sungkyunkwan University, Seoul, Korea, 3 Department of Immersive Media Engineering, Sungkyunkwan University, Seoul, Korea

☯ These authors contributed equally to this work.
* alohakim@skku.edu

**Data Availability Statement:** All data files employed in the experiment are available from the

## Abstract

This study presents a comprehensive exploration of topic modeling methods tailored for large language model (LLM) using data obtained from Web of Science and LexisNexis from June 1, 2020, to December 31, 2023. The data collection process involved queries focusing on LLMs, including "Large language model," "LLM," and "ChatGPT." Various topic modeling approaches were evaluated based on performance metrics, including diversity and coherence. latent Dirichlet allocation (LDA), nonnegative matrix factorization (NMF), combined topic models (CTM), and bidirectional encoder representations from Transformers topic (BERTopic) were employed for performance evaluation. Evaluation metrics were computed across platforms, with BERTopic demonstrating superior performance in diversity and coherence across both LexisNexis and Web of Science. The experiment result reveals that news articles maintain a balanced coverage across various topics and mainly focus on efforts to utilize LLM in specialized domains. Conversely, research papers are more concise and concentrated on the technology itself, emphasizing technical aspects. Through the insights gained in this study, it becomes possible to investigate the future path and the challenges that LLMs should tackle. Additionally, they could offer considerable value to enterprises that utilize LLMs to deliver services.

## Introduction

The advent of large language model (LLM) marks a significant moment in the domain of artificial intelligence and natural language processing (NLP). Specifically, the development of models such as generative pre-trained Transformer (GPT) series has brought a significant shift, completely changing how machines comprehend and generate natural language [1].

LLMs represent the advancement of deep learning, especially by employing transformer architectures to enhance their ability to process and comprehend extensive textual data [2].

Github database (URLs: https://github.com/
Haein34/LLM-with-Topic-modeling/tree/main).

**Funding:** This study was supported by a National
Research Foundation of Korea (NRF) (http://nrf.re.
kr/eng/index) grant funded by the Korean
government (RS-2023-00208278). The funders
had no role in study design, data collection and
analysis, decision to publish, or preparation of the
manuscript.

**Competing interests:** The authors have declared
that no competing interests exist.

The rapid progress of this groundbreaking technology has integrated into everyday life, providing assist while also introducing new challenges and controversies [3].

The enormous scale and complexity of LLMs raise concerns about ethical considerations, embedded biases within data, environmental impacts due to extensive computations, and the broad societal implications derived from the application of these models [4]. Additionally, despite the remarkable capabilities, LLMs have limitations, such as challenges in interpretability and potential adversarial vulnerabilities [5].

However, among these concerns lies the importance and potential of LLMs. Their ability to grasp contextual knowledges, produce consistent text, and adapt to diverse linguistic styles holds tremendous promise across various domains [6–11]. The influence of LLMs, contributing to the advancement of fields like healthcare and education by aiding language translation, information retrieval, automation, and content generation, is extensive. Therefore, leveraging the potential of LLMs while addressing challenges and conducting research to explore their possibilities is significantly imperative.

This research aims to enrich the body of knowledge by employing data mining with unsupervised machine learning methods to recognize and derive insights from LLM-related text within two unique platforms: LexisNexis representing the media and Web of Science representing academia, contributing to perception and insight into the field. Ultimately, this study discovers the unique meanings inherent within LLM and enhance both commercial and academic value. The significance of such analysis lies in its ability to provide value as feedback within industries relying on LLMs, thereby facilitating the exploration of new insights. Additionally, the insight from the experiment can be utilized when adjusting relevant product or service strategies.

## Related works

### Research on utilization of LLMs in various domain

As LLM-related industry experienced rapid growth and a surge in interest, related research is being actively conducted in various fields.

[7] introduced the evolution of LLM applications and explored their current utilization in medical environments. The authors assessed the capabilities and limitations of LLM and examined the potential to improve the efficiency of work in medicine across clinical, educational, and research domains. [8] proposed an approach to utilize LLM in learning environments and educational curriculum. In addition, the authors investigated the risks of immediate challenges such as potential bias and potential for misuse. To suggest the possibility of LLM being applied across various domain [9], performed trend and distribution analysis. The results implemented that LLM can be applied in a variety of areas, from physics and mathematics to history and education. In addition, efforts to apply LLM continue not only in various fields, but also in specialized domains requiring expertise, such as the medical domain. [10] developed benchmark data to apply LLM to actual clinical applications and attempted to achieve encouraging performance by introducing prompt tuning. [11] released the Lawformer model, a pre-trained language model based on Longformer, to provide legal systems through NLP. This model was created with the understanding that legal documents necessitate the processing of significantly longer tokens compared to general documents.

In summary, these studies have underscored the adaptability of LLMs, affirming their ability to be widely utilized in diverse field and demonstrating the potential benefits of employing LLMs across diverse domains such as problem-solving, automation, and natural language comprehension.

## Topic modeling

Topic modeling, a statistical model employed in NLP, aims to identify main themes, known as topics, within document collections [12]. In other words, it is a text mining technique that is utilized to uncover hidden semantic structures within textual data, and finds applications across diverse fields.

[13] performed a comparative study of different topic models, encompassing latent Dirichlet algorithm (LDA), correlated topic model, hierarchical Dirichlet process (HDP), and Dirichlet multinomial regression (DMR). The authors employed adolescent drug use and depression as queries for their analysis. [14] employed LDA, nonnegative matrix factorization (NMF), and bidirectional encoder representations from Transformers topic (BERTopic) on Twitter posts, performing a comparative assessment for each individual topic modeling algorithm. [15] harnessed BERTopic embeddings and class-based term frequency inverse document frequency (c-TF-IDF) to created dense clusters, conducting an analysis of the top themes associated with disparities in cancer care and the model was evaluated using topic coherence scores. [16] experimented with comparing the coherence values using LDA and NMF on literature related to coronavirus disease (COVID-19) to determine which topic model produces a higher-quality representation. [17] utilized LDA analysis on Twitter data to assess the efficiency of the energy market. Subsequently, the derived topic was employed in a classification model to evaluate the accuracy of predicting market movements. [18] used BERTopic on environmental, social, and governance text data to analyze comprehensive evaluation. [19] employed BERTopic to analyze text data on Bitcoin from LexisNexis, Web of Science, and Reddit, representing the media, academia, and public. Additionally, dynamic topic modeling (DTM) was used to monitor changing themes overtime. As illustrated, topic modeling has aided in extracting clusters and meanings from large text datasets. Consequently, researchers across various fields were able to comprehend and analyze the main themes and insights within large document collections, thus demonstrating the applicability of topic modeling across diverse domains.

Despite the rapid growth of LLMs and their wide application in diverse domains, there is a complete absence of research regarding topic modeling specifically designed for LLMs. Therefore, analyzing and understanding text related to LLMs, along with attaining insights and perceptions, would be an essential step for the development and growth of LLMs.

## Methods

The entire experimental flow diagram is described in Fig 1.

## Data collection

To collect as diverse data as possible, data was gathered through queries related to LLM and the name of prominent LLM technology. Specifically, the goal was to search and collect all the searchable fields containing the query "Large language model," "LLM," and "ChatGPT" as search terms. While there are numerous LLMs beyond ChatGPT, to reduce noise and derive significant results, only the most representative model name was utilized.

Gathering data from LexisNexis and Web of Science, the authors aimed to understand the perceptions toward LLM across news and academic. In detail, the LexisNexis data was collected by accessing Lexis+ and subsequently using a Python scraper to gather.DOCX format files, which were then preprocessed into comma-separated values format. For the Web of Science data, downloads were performed for.XLSX files. The access to LexisNexis database data may require an institutional paid subscription. Specifically, LexisNexis provided complete

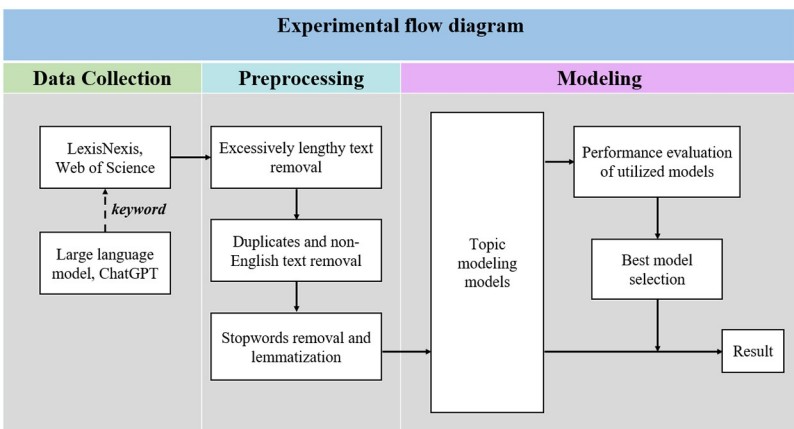

**Fig 1. Experimental diagram.**

news articles from major news publications over the world and Web of Science contributed academic paper abstracts.

The collected data spans from June 1, 2020 to December 31, 2023, and includes 10,563 newspapers, and 11,070 academic papers.

## Data preprocessing

The collected texts went through a process to eliminate duplicates. Additionally, data exceeding certain length thresholds were regarded as noise and consequently removed. Following this, the spaCy library was employed to perform lemmatization aimed at isolating nouns from the data while eliminating irrelevant words through stop words removal [20]. After a series of preprocessing steps, any data lacking textual content was removed. Consequently, the experiment employed a total of 3,917 texts sourced from LexisNexis and 3,438 texts from Web of Science.

## Topic analysis based on performance evaluation metrics

Unsupervised topic modeling methods and the nature of NLP dictate that the optimal approach can vary depending on the text data being analyzed [21, 22]. Therefore, the authors initially assessed the most suitable method for each platform under analysis through evaluation metrics and subsequently conducts experiment.

**Evaluation metrics.** The authors employed two evaluation metrics for comparing topic modeling method's performance.

The first evaluation metric is topic diversity. The diversity evaluates the range of topics included, referring in topic modeling to the extent to which each topic represents different subjects or concepts. Topic diversity ranges from 0 to 1 and is one of the important factors in evaluating the quality of a topic model [23]. According to [24], an excessively large number may lead to topics that share commonalities, with overlapping key queries. Conversely, an excessively small number may lead to complex and expansive topics that pose challenges in interpretation. Therefore, if the diversity falls within the range of 0.7 to 0.9, it can be considered an appropriate diversity value.

Topic equation for topic diversity can be stated as follows:

$$Topic\ Diversity = \frac{M}{N \times k} \tag{1}$$

Where *M* represents the aggregate count of unique words extracted across all topics. *N* denotes the total number of topics in the model output, and *k* signifies the count of top words extracted from each individual topic.

The second evaluation metric is topic coherence value. The coherence value examines how well words within a topic relate in meaning [25]. Therefore, achieving a higher coherence value is desirable for connecting similar words within each topic group. Among diverse coherence measures, the authors selected for *c_v* based on the advantage presented in [26, 27]. *c_v* uses a sliding window of size 110 to calculate the co-occurrence count of the given words. This count is used to compute the normalized pointwise mutual information (NPMI) between all keywords, resulting in a set of vectors for each key word. The one-set segmentation of primary queries calculates the similarity between each key query vector the aggregate of all keyword vectors. Consequently, cosine similarity is used for this similarity measurement and Coherence is the average of these similarities. The equation for NPMI is expressed as follows:

$$NPMI\left(w_i,\ w_j\right)^{\gamma} = \left(\frac{PMI\left(w_i,\ w_j\right)}{-\log P(w_i,\ w_j) + \epsilon}\right)^{\gamma} \tag{2}$$

$$PMI\left(w_i,\ w_j\right) = log\frac{P\left(w_i,\ w_j\right) + \epsilon}{P(w_i) \cdot P(w_j)} \tag{3}$$

The probability is calculated through the consideration of the frequency of word co-occurrences, denoted as $w_i$ and $w_j$. $\epsilon$ represents a smoothing count value implemented to avoid the numerator from reaching zero and $\gamma$ determines the weight of NPMI.

Elevated coherence implies that the words in a topic share cohesive semantic connection, enhancing the interpretability and meaningfulness of the uncovered themes. Additionally, comparing the coherence values within the same model enabled estimating the optimal number of topics.

**Topic modeling models.** The first topic modeling algorithm employed is LDA. LDA is a statistical model for topic modeling that assumes a combination of topics for a document collection and models how each document is distributed across these topics [28]. The LDA is widely used to derive topics from text data [29]. The time complexity of the LDA based on Gibbs sampling and can be expressed as follows [30].

$$Time\ complexity\ (LDA) = O(N \cdot M \cdot K \cdot iter\#) \tag{4}$$

In the equation, *N* and *M* refer to the number of documents and words, *K* is the number of topics, and *iter#* is the number of repetitions.

The second topic modeling method utilized is NMF. NMF involves representing a nonnegative matrix as the result of multiplying two nonnegative matrices of lower rank [31]. When applied to term frequency-inverse document frequency transformed data, NMF breaks down the matrix into two smaller matrices. Due to its ability to provide semantically significant results in clustering tasks, NMF has become popular as both a segmentation and topic modeling technique, particularly in document analysis [32]. The time complexity of NMF is as follows.

$$Time\ complexity\ (NMF) = O(n^2 \cdot m \cdot k) \tag{5}$$

In the equation, $n^2$ is the number of documents corresponding to the number of rows of

the data matrix, *m* is the number of words corresponding to the column, and *k* is the number of topics [33].

The third methodology applied is combined topic model (CTM). CTM refers to composite topic modeling approaches that combine the bag of words with sentence BERT (SBERT), a document representation technique from pre-trained language models [34]. CTM aims to derive topics from documents by employing the contextual expression of BERT's document embeddings and merging them with the unsupervised learning abilities inherent in conventional topic models. Since the time complexity of CTM depends on the BERT model, it depends on the input token $O(n)$. However, since the CTM incorporates the LDA with products of experts (ProdLDA) model with BERT, the time complexity stemming from Gibbs sampling is considered together.

The last algorithm utilized is BERTopic. BERTopic utilizes BERT embeddings alongside c-TF-IDF to create compact clusters that preserve essential words within topics [35]. Furthermore, BERTopic's modular structure facilitates the integration and utilization of various algorithms. By employing contextual word and sentence vector representations, BERTopic leverages semantic properties to position similar texts closely in the vector space. It leverages SBERT for document embedding, which generates dense vectors for sentences and paragraphs. Dimensionality reduction is then executed using uniform manifold approximation and projection (UMAP), followed by cluster creation via hierarchical density-based spatial clustering of applications with noise (HDBSCAN) [18]. Prior to computing c-TF-IDF, the authors first utilized CountVectorizer to generate topic representation. This preprocessing step is intended to reduce the dimensions of the c-TF-IDF matrix. Subsequently, topic representation is obtained from each cluster using c-TF-IDF. In this approach, each clustered document is treated as a single document, with inverse document frequency substituted by inverse class frequency to assess the relevance of words for the class. This methodology reinforces the importance of words within a cluster, leading to the creation of a topic-word distribution for each document cluster. This procedure simplifies the process of identifying representative words for the respective topics. The time complexity of BERTopic is $O(n)$, reflecting the complexity of BERT.

## Experiment

### Evaluation metric comparison for platform-specific topic modeling models

The authors computed evaluation metrics for each analysis target platform across all topic models and compared the results (Fig 2, Table 1). Through this process, the validity of the employed unsupervised topic modeling utilized in the respective platform has been ensured.

In summary, BERTopic demonstrated superior overall performance across LexisNexis and Web of Science, excelling in diversity and coherence values. Consequently, the BERTopic was utilized in the experiment. Additionally, the coherence value was calculated based on different topic numbers for verifying the optimal number of topics on each target platform (Figs 3 and 4). The hyperparameters utilized for the experiment is described in Table 2. For SBERT, 'multi-qa-miniLM-L6-cos-v1' was employed based on the comparison experiment results from [36].

### Results of the topic analysis for LexisNexis

In this section, the authors present the outcomes derived from applying the BERTopic model to the textual data acquired from LexisNexis. To characterize and label the identified topics, the top 8 words were extracted for each topic, as depicted in Table 3. Then the main theme of each topic was labeled by authors derived from the top keywords with the highest importance

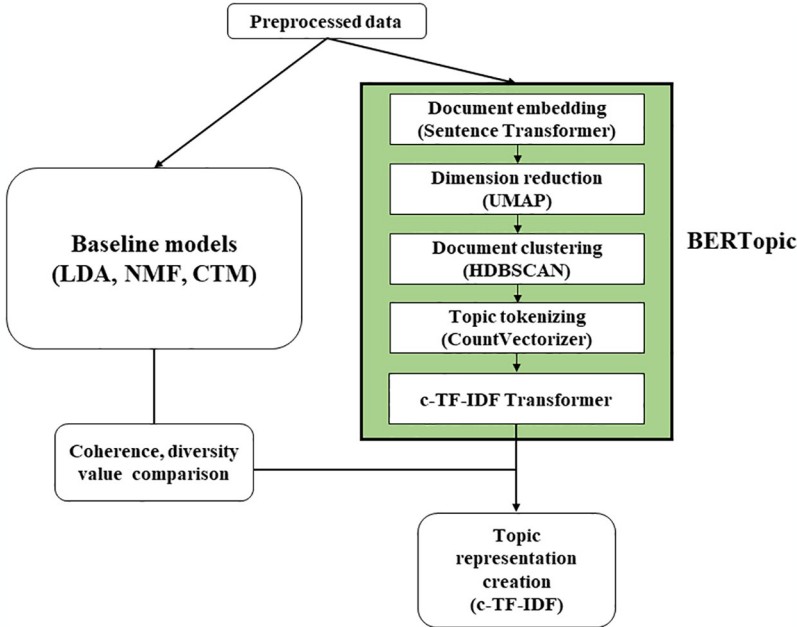

**Fig 2. Detailed schematic diagram of the comparison of evaluation metrics between models.**

included in the topic and from the original data. This method is predominantly used in papers utilizing topic modeling algorithms [37–39].

Topic 0 was described as "Possibility of using artificial intelligence (AI) technology in a wide range of fields" through "ai," "chatgpt," "technology," "student," "use," "chatbot," "work," and "human." AI technology, including LLM, is being used in a wide range of areas. [40] argued that AI tools such as Copliot, based on LLM, have brought improved productivity in human life. These tools typically emphasized meaningful increases in speed of execution without significantly compromising quality in research. [41] proposed that LLM can be utilized to help students accomplish their academics. To prove this, the authors conducted in-depth case studies to measure the quality of LLM as a learning tool.

Topic 1 was denoted as "Spread of graphics processing unit (GPU) market and LLM company" through "company," "chip," "nvidia," "market," "openai," and "chatgpt." The era has arrived where the development of new AI models demands increased memory and extensive

**Table 1. Statistical comparison of evaluation metrics among topic models.**

| Target platform | Method | Diversity | | | Coherence values (c_v) | | |
|---|---|---|---|---|---|---|---|
| | | Mean | Median | Standard deviation | Mean | Median | Standard deviation |
| LexisNexis | LDA | 0.6772 | 0.6750 | 0.0709 | 0.3280 | 0.3032 | 0.0789 |
| | NMF | 0.3684 | 0.3750 | 0.0616 | 0.2366 | 0.2377 | 0.0140 |
| | CTM | 0.8944 | 0.8450 | 0.0781 | 0.3864 | 0.3928 | 0.0443 |
| | BERTopic | 0.8579 | 0.8610 | 0.0902 | 0.4544 | 0.4462 | 0.0268 |
| Web of Science | LDA | 0.6891 | 0.6931 | 0.0677 | 0.3493 | 0.3644 | 0.0592 |
| | NMF | 0.3914 | 0.4167 | 0.0859 | 0.2892 | 0.2934 | 0.0117 |
| | CTM | 0.8961 | 0.8450 | 0.0767 | 0.3640 | 0.3673 | 0.0415 |
| | BERTopic | 0.9047 | 0.8840 | 0.0329 | 0.4389 | 0.4552 | 0.0413 |

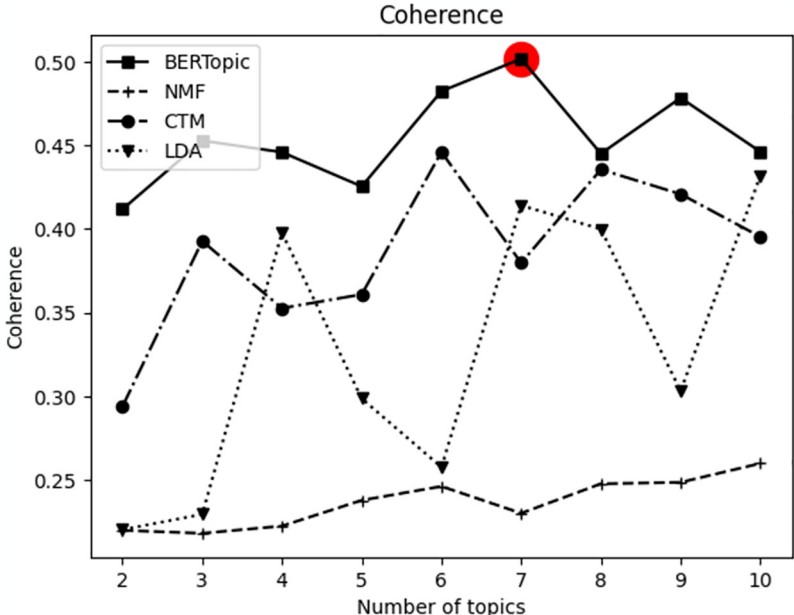

**Fig 3. Comparison of coherence values based on the number of topics from the LexisNexis data.**

datasets. To meet this requirement, AI chips are continually advancing, with companies like OpenAI consistently investing in larger memory and resources to introduce sophisticated models [42]. Specifically, as the functionality of baseline models improves, their sizes are advancing at a rate surpassing Moore's Law. The GPT-4 model, projected to be the largest by 2023, is estimated to range between 200 billion and 300 billion parameters, necessitating

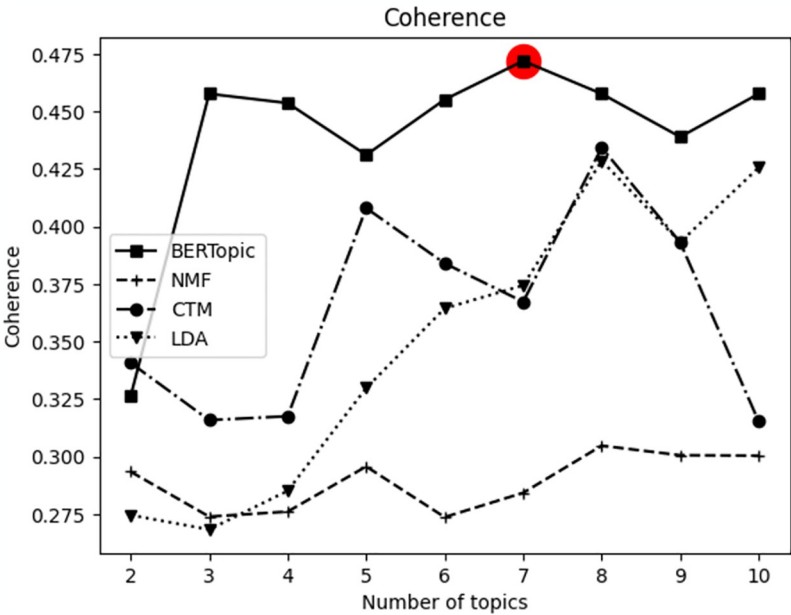

**Fig 4. Comparison of coherence values based on the number of topics from the Web of Science data.**

**Table 2. The hyperparameters used in the experiment.**

| Package version | | | | | |
|---|---|---|---|---|---|
| joblib = = | octis = = | Bertopic = = | sklearn = = | umap = = | genism = = |
| 1.1.0 | 1.13.1 | 0.16.0 | 1.1.0 | 0.5.6 | 4.2.0 |

| Hyperparameters | | | |
|---|---|---|---|
| Function / Platform | SentenceTransformer | UMAP | HDBSCAN |
| LexisNexis | 'multi-qa-miniLM-L6-cos-v1' | n_neighbors = 30, n_components = 3, min_dist = 0.2, metric = 'cosine' | min_cluster_size = 16, metric = 'euclidean', cluster_selection_method = 'eom' |
| Web of Science | | n_neighbors = 20, n_components = 3, min_dist = 0.15, metric = 'cosine' | |

substantial computing power for training [43]. Consequently, the increasing market interest in the new AI chip was noticeable.

Topic 2 was labelled as "LLM for social media" through "twitter," "thread," "user," "musk," "zuckerberg," "platform," "instagram," "linkedin." Social media platforms like Twitter and Reddit have leveraged AI models to enhance user convenience. This application has further expanded with the emergence of LLMs. A study by [44] investigated whether LLM-generated text differs from human-generated text and discovered previously unknown campaigns on platforms like Twitter. Another study [45] examined fake profiles created through LLMs on the LinkedIn social network. Detecting fake profiles is crucial for maintaining platform integrity and improving confidence against potential phishing and fraud activities. With the introduction of LLMs into social media, users are exposed to new aspects, and various perspectives are being presented.

Topic 3 was nominated as "LLM in legal cases" through "case," "legal," "lawyer," "court," "judge," "schwartz," "cohen," and "decision." As the scope of use of LLM increases, attempts to

**Table 3. Topic representations of LLM from LexisNexis data.**

| Topic | Main theme | Document count | Top keywords with the highest importance |
|---|---|---|---|
| 0 | Possibility of using AI technology in a wide range of fields | 1,678 | ('ai', 0.0421) ('chatgpt', 0.0294) ('technology', 0.0216) ('student', 0.0196) ('use', 0.0188) ('chatbot', 0.0182) ('work', 0.0167) ('human', 0.016) |
| 1 | Spread of GPU market and LLM company | 934 | ('ai', 0.0444) ('company', 0.0346) ('chip', 0.0229) ('nvidia', 0.0224) ('market', 0.0217) ('openai', 0.0209) ('chatgpt', 0.0193) ('tech', 0.0188) |
| 2 | LLM for social media | 43 | ('twitter', 0.0963) ('thread', 0.0931) ('user', 0.0713) ('musk', 0.0541) ('zuckerberg, 0.0453) ('platform', 0.0441) ('instagram', 0.0418) ('linkedin, 0.04) |
| 3 | LLM in legal cases | 41 | ('case', 0.0832) ('legal', 0.0794) ('lawyer', 0.0766) ('court, 0.0702) ('judge', 0.0603) ('schwartz', 0.057) ('cohen', 0.0336) ('decision', 0.0307) |
| 4 | Youth seeking assistance from AI chatbots | 20 | ('young', 0.1246) ('people, 0.0917) ('sex', 0.0701) ('onlin', 0.0609) ('nominet', 0.0501) ('medical', 0.0425) ('chatbot', 0.0421) ('ai', 0.0399) |
| 5 | Altman's World Coin launch | 18 | ('worldcoin, 0.2125) ('scan', 0.0994) ('bitcoin', 0.0793) ('crypto', 0.0733) ('token', 0.0658) ('altman', 0.0588) ('cryptocurrency', 0.0523) ('orb', 0.0455) |
| 6 | AI chatbot used in the Russia-Ukraine war | 17 | ('russia', 0.0858) ('russian', 0.0744) ('putin', 0.0534) ('korotkov', 0.0439) ('fr', 0.0439) ('train', 0.0399) ('frm', 0.0288) ('ukraine', 0.0279) |

utilize it for legal exploration and scenario application continue. [46] highlighted that the complexity and ambiguity of legal provisions lead to inconsistent application by lawyers. In such scenarios, the authors proposed a new technique called 'chain of reference' using LLMs to suggest a framework with improved performance for answering legal questions. However, there were instances where this usage was not yet justified. For example, Donald Trump's attorney, Michael Cohen, submitted fake case quotes to his attorney (i.e., Schwartz) in official court filings without realizing that the quotes generated by Google Bard were fictitious [47]. In the aftermath of such cases, courts nationwide were reported to be struggling with how to regulate the use of generative AI programs like ChatGPT.

Topic 4 was named "Youth seeking assistance from AI chatbots" through "young," "people," "sex," "online," "nominet," "medical," "chatbot," and "ai." Many young people are using AI chatbots like ChatGPT to seek assistance with their studies, emails, and work. According to a survey conducted by Nominet in the UK, 53% of respondents reported using AI chatbots. Nominet CEO Paul Fletcher stated that the acceptance of technology among young people and its integration into their daily lives were encouraging [48]. In another example [49], suggested that the development of chatbots could be utilized to deal with the mental wellness of children and adolescents. Additionally, the authors expressed caution regarding potential bias or misinformation.

Topic 5 was stated "Altman's World Coin launch" through "worldcoin," "bitcoin," "crypto," "token," "altman," "cryptocurrency," and "orb." Sam Altman, the founder of ChatGPT, launched World Coin with the aim of establishing a blockchain-based identity and financial network. This process includes individuals scanning their irises using a device called an orb, verifying their uniqueness as human entities, and subsequently being rewarded with World Coin tokens. Altman initiated this project to facilitate global economic access for everyone [50].

Topic 6 was termed "AI chatbot used in the Russia-Ukraine war" through "russia," "putin," "korotkov," and "ukraine." After ChatGPT became the latest trend on social media, it found application in discussions about the Russia-Ukraine war. In October 2023, the Indian Foreign Minister requested a negotiation blueprint for the Russia-Ukraine war, and ChatGPT responded with eight solutions [51]. Shashi Tharoor, a senior leader of India's Congress, also took notice and argued that although the role of a leader in a conflict situation may differ from what AI claims, it is a valuable experiment [52]. Therefore, it is essential to recognize that ChatGPT can be used politically, and research into hidden biases and risks must continue [53].

The visualization in Figs 5 and 6 confirmed both the similarity matrix and intertopic distances. Fig 5 is a heatmap that represents topic embeddings as vectors and measures the similarity between two topics by calculating the cosine similarity between each vector [26, 54]. And Fig 6 is the intertopic distance plot, which utilizes UMAP to reduce topic embedding vectors to two dimensions (2D) and considers the similarity between topic embeddings using cosine distance [26, 54]. In the 2D UMAP projection, the x-axis and y-axis represent the dimensionally reduced representations of the data. Subsequently, the circles are placed based on the distances between topics, and the size of each circle reflects the relative size of each topic, thus indicating the relative importance of each topic. Through those visualization, the relative importance of each topic can be easily identified and visually understood. The authors could examine the similarity between topics through the heatmap in Fig 5.

Meanwhile, in Fig 6, besides the two main themes, the remaining of the topics demonstrate a consistent, uniform distribution. These visualization outcomes indicate that news articles tend to cover a broad spectrum of topics of LLMs.

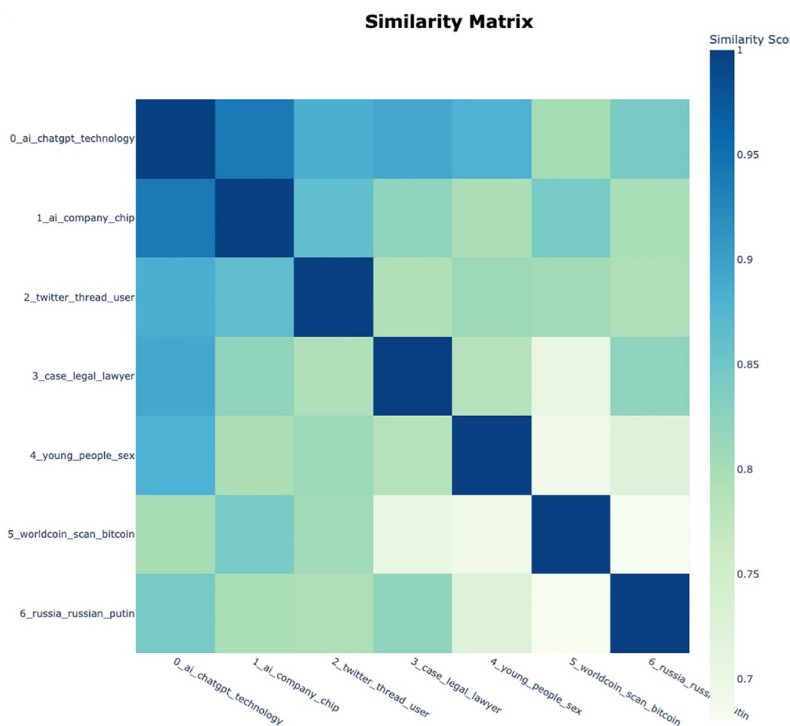

**Fig 5. Similarity matrix derived from topic modeling results on LexisNexis data.**

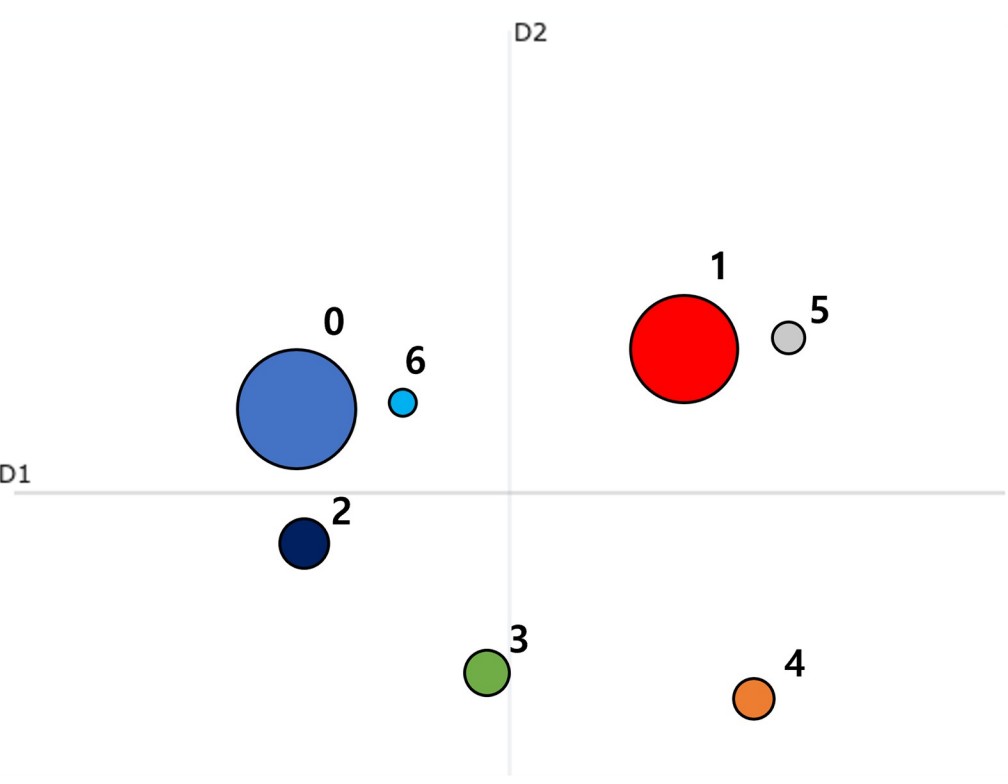

**Fig 6. Distance between topics from topic modeling results on LexisNexis data.**

Table 4. Topic representations of LLM from Web of Science data.

| Topic | Main theme | Document count | Top keywords with the highest importance |
|---|---|---|---|
| 0 | Utilization of LLM across various domain | 1,146 | ('chatgpt', 0.0259) ('ai', 0.0205) ('llm', 0.0180) ('learning', 0.0174) ('knowledge', 0.0147) ('processing', 0.0145) ('natural', 0.0142) ('application', 0.0141) |
| 1 | Research on utilization of LLM in the field of education and learning | 310 | ('student', 0.0481) ('school', 0.0257) ('english', 0.0253) ('teacher', 0.0210) ('learning', 0.0197) ('teaching', 0.0195) ('factor', 0.0191) ('personality', 0.0182) |
| 2 | Research on LLM for utilizing voice recognition | 239 | ('speech', 0.0619) ('effect', 0.0309) ('acoustic', 0.0292) ('speaker', 0.0291) ('bilingual', 0.0256) ('recognition', 0.0234) ('auditory', 0.0211) ('voice', 0.0196) |
| 3 | Research on tuning LLM for improving the efficiency | 161 | ('high', 0.0241) ('time', 0.0227) ('parameter', 0.0197) ('structure', 0.0196) ('simulation', 0.0191) ('flow', 0.0187) ('observation', 0.0170) ('field', 0.0169) |
| 4 | The utilization of LLM for disease and risk detection in the medical field | 127 | ('child', 0.0462) ('outcome', 0.0361) ('age', 0.0329) ('clinical', 0.0326) ('risk', 0.0315) ('group', 0.0280) ('treatment', 0.0274) ('participant', 0.0246) |
| 5 | The utilization of LLM in the field of traffic and transportation | 56 | ('accident', 0.0510) ('management', 0.0406) ('safety', 0.0398) ('road', 0.0335) ('traffic', 0.0278) ('incident', 0.0255) ('country', 0.0252) ('development', 0.0244) |
| 6 | Highlighting LLMs' potential for assisting gesture recognition | 32 | ('sign', 0.2437) ('gesture', 0.1103) ('hand', 0.1015) ('recognition', 0.0912) ('signer', 0.0467) ('image', 0.0425) ('slr', 0.0344) ('accuracy', 0.0343) |

## Results of the topic analysis for Web of Science

The results obtained from employing the BERTopic on the textual data sourced from Web of Science are illustrated in this section. For each main theme, the top 8 keywords with the utmost importance were extracted to determine and assign names to the identified topics (Table 4). The naming process of main topics followed the same approach used for naming themes in LexisNexis.

Topic 0 was classified as a broad category, representing a wide-ranging theme of "Utilization of LLM across various domain." Topics 1 to 6 appeared to have originated from this primary category.

Topic 1 was denoted as "Research on utilization of LLM in the field of education and learning" through "student," "school," "english," "teacher," "learning," and "teaching." There is active progress in studies aiming to utilized LLM for education. [8] explored the present uses of LLM in education, covering opportunities and challenges from both teacher and student perspectives. [55] considered methods to integrate ChatGPT into language courses, along with the potential advantages and obstacles that could arise from this integration.

Topic 2 was named "Research on LLM for utilizing voice recognition" through "speech," "acoustic," "speaker," "recognition," "auditory," and "voice." Research on models that go beyond simple text based LLM to utilize multimodal approaches is actively being conducted. [56] conducted research on the possibility of directly attaching an audio encoder to LLMs to convert them into automatic speech recognition (ASR) systems, allowing their text counterparts to be used in the exact same manner. It has also been confirmed that when combining LLM with prompt engineering and fine-tuning, they can function as post-recognition processors for speech, conducting revision and error adjustment [57].

Topic 3 was labelled as "Research on tuning LLM for improving the efficiency" through "high," "time," "parameter," "structure," "simulation," and "flow." The success of LLMs like ChatGPT and GPT-4 has facilitated cost-effective testing. Various studies have been introduced to efficiently adjust parameters. [58] proposed a method for tuning external parameters of LLMs called adapter-based parameter-efficient fine-tuning (PEFT). Considering this, the authors designed the LLM-adapters framework, enabling the integration of various adapters to perform tasks. [59] outlined an efficient strategy to transform LLMs into multimodal LLMs. Specifically, methodology was adopted to reduce trainable parameters by adjusting the

LayerNorm within each attention block. All of these methods aim to efficiently manage parameters, reducing training time and facilitating effective LLM learning.

Topic 4 was defined as "The utilization of LLM for disease and risk detection in the medical field" through "clinical," "risk," "treatment," and "participant." Research endeavors aiming to utilize LLMs in clinical settings are actively underway. [7] explored use cases in medicine considering the potential to enhance the productivity of clinical, academic, and research tasks by considering the advantages and limitations of LLMs. [60] conducted a performance evaluation of ChatGPT in medical exams, demonstrating that LLMs may offer assistance in medical education and potential clinical decision-making.

Topic 5 was identified as "The utilization of LLM in the field of traffic and transportation" through "accident," "management," "safety," "road," "traffic," and "development." [61] demonstrated the use of ChatGPT to address critical transportation safety issues. Additionally, the authors discussed controversies surrounding LLM and proposed solutions. Ultimately, the authors argued that LLM will shape components of traffic safety research and potentially foster advancement in this field. [62] created and evaluated an LLM-based framework to uncover underreported crash factors by analyzing accident narratives. The authors investigated the identification accuracy of various LLM frameworks, emphasizing the potential of an LLM-based framework to effectively identify and address underreported collision factors.

Topic 6 was termed "Highlighting LLMs' potential for assisting gesture recognition" through "sign," "'gesture," "hand'," "recognition," "signer," "image," "'slr," and "accuracy." [63] employed LLM to understand American sign language through robots and illustrates the capability of this approach in real scenarios to facilitate non-verbal human-robot interaction. [64] explored the potential of using LLM to conduct experiments, investigating the possibilities of inferring hand gestures through tracking light and vibration sensor data. As a result, the authors emphasized the potential of LLM in sensor data analysis for gesture recognition.

The analysis of Web of Science data showed that the main emphasis in research papers is on employing and improving the efficiency of LLM in diverse fields.

In addition, the similarity matrix was confirmed through visualization in Fig 7. Topic 0 exhibited significant similarity with all other topics, potentially suggesting a comprehensive topic that encompasses all other topics. Moreover, it was observed that the remaining topics exhibit a balanced distribution of similarity among themselves.

Fig 8 illustrates the distribution of distances between attained topics. There was an appropriate distance between topics without any overlap, and it was evident that Topic 0 contains a significantly larger quantity of data compared to the other topics. These visualizations suggest that, unlike news articles, the research paper derives from a single comprehensive theme and is topic-focused. Additionally, statistical analysis of the similarity between topics were performed on both platforms (Table 5).

## Conclusion

Despite the swift expansion and extensive use of LLMs across various fields, there remains a significant gap in research concerning topic modeling conducted specifically for LLMs. Consequently, examining and comprehending text associated with LLMs, while investigating insights and perceptions, represents an essential step towards promoting the progress and growth of LLMs.

This study gathered unstructured text data from LexisNexis and Web of Science, representing media and academia, respectively. The data used in the experiment spans from June 2020 when GPT-3 was first released to January 2024. Additionally, the authors first compared the

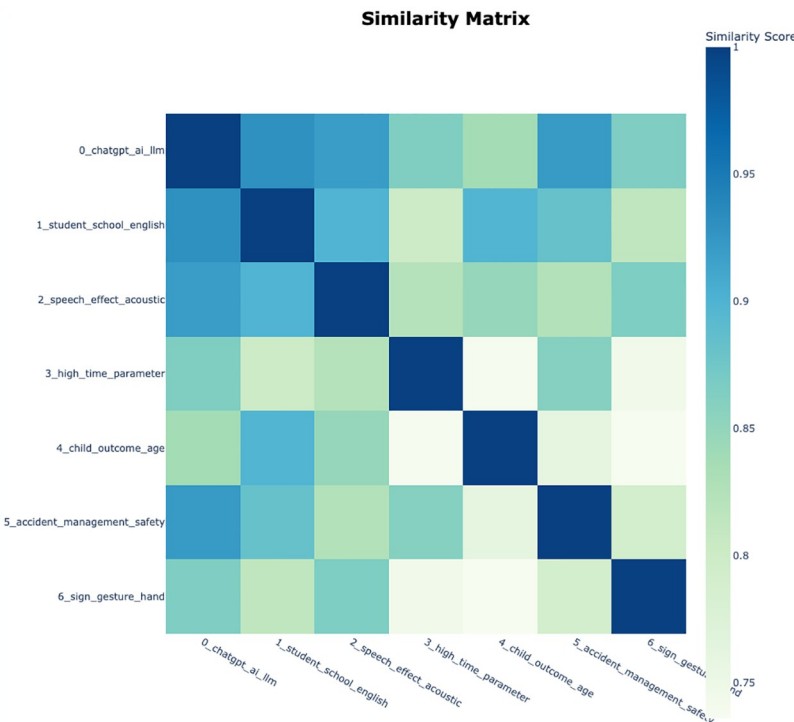

**Fig 7. Similarity matrix derived from topic modeling results on Web of Science data.**

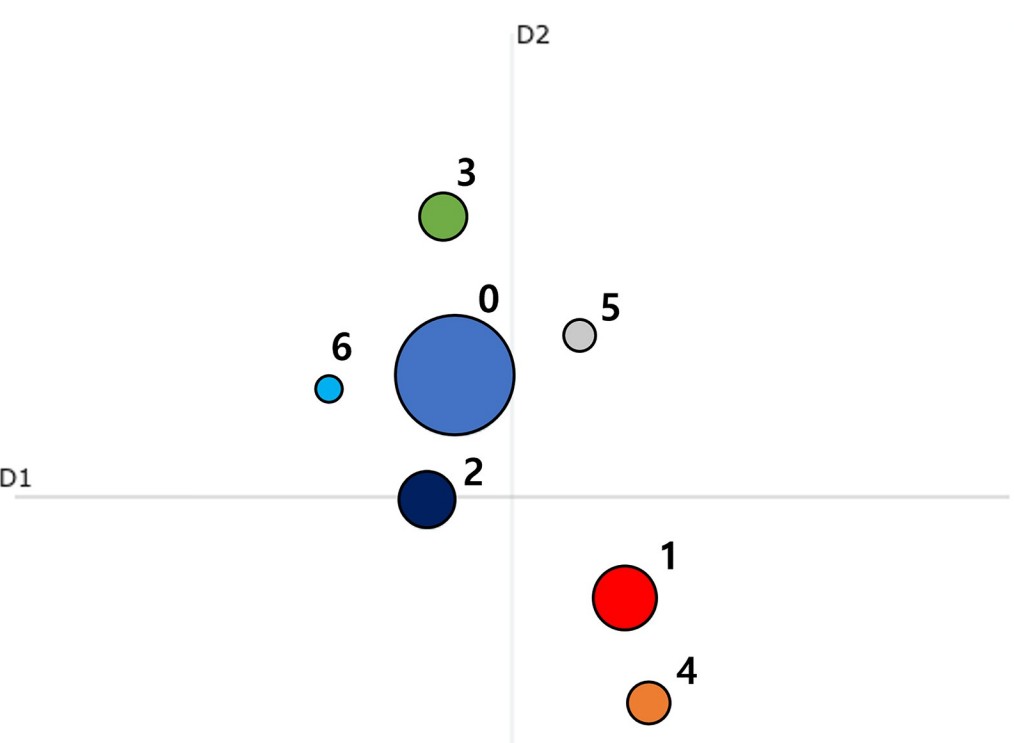

**Fig 8. Distance between topics from topic modeling results on Web of Science data.**

**Table 5. Statistical analysis of the similarity between topics on LexisNexis and Web of Science data.**

| Platform | Similarity between topics | | |
|---|---|---|---|
| | Mean | Median | Standard deviation |
| LexisNexis | 0.8466 | 0.8583 | 0.0593 |
| Web of Science | 0.8082 | 0.8054 | 0.0652 |

evaluation metrics of four topic modeling methodologies to identify and utilize the optimal methodology for each target platform.

These results have the following theoretical implications. First, media outlets such as news increase public awareness by emphasizing the application of LLM. This promotes public understanding and interest in the professional field [65]. Second, an emphasis on technical aspects in academia provide valuable information to experts in the field and play an important role in technological advancement. Third, the balance between diversity and technical depth between news articles and research papers maintains harmony between expertise and public consciousness, which contributes to effective communication and knowledge transfer [25].

The practical implications can be summarized as follows. This study has important value for companies that provide services by applying LLM. For instance, companies serving the general public should ensure a range of diverse services. Conversely, enterprises integrating LLM with specialized fields like academia should prioritize offering sophisticated and advanced technologies.

Consequently, the findings from the experiment demonstrate that news articles provide even coverage of diverse topics, emphasizing the application of LLM in specialized fields. In contrast, research papers are more compact, concentrating primarily on the technology itself and emphasizing technical aspects.

However, this study carries several limitations for future exploration. This research exclusively processed English-language data, while countries exhibit diverse LLM related perceptions and text. Furthermore, subsequent research could investigate public interest from social media text data, which was not covered in this study. In addition, further comparative analysis with emerging deep learning-based topic modeling methods should be supplemented in future studies.

## Acknowledgments

We would like to thank Editage (https://www.editage.co.kr) for the English language editing.

## Author Contributions

**Conceptualization:** Hae Sun Jung, Haein Lee, Jang Hyun Kim.

**Data curation:** Hae Sun Jung, Haein Lee.

**Formal analysis:** Hae Sun Jung, Haein Lee, Young Seok Woo.

**Investigation:** Hae Sun Jung, Haein Lee.

**Methodology:** Hae Sun Jung, Haein Lee, Young Seok Woo, Seo Yeon Baek.

**Validation:** Hae Sun Jung, Haein Lee.

**Visualization:** Hae Sun Jung, Haein Lee.

**Writing – original draft:** Hae Sun Jung, Haein Lee, Jang Hyun Kim.

**Writing – review & editing:** Hae Sun Jung, Haein Lee, Jang Hyun Kim.

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
