## [Decision Letter · Decision Letter 0]

1 Apr 2024

PONE-D-24-00612Expansive Data, Extensive Model: Investigating LLM through Unsupervised Machine Learning in Academic Papers and NewsPLOS ONE

Dear Dr. Kim,

Thank you for submitting your manuscript to PLOS ONE. After careful consideration, we feel that it has merit but does not fully meet PLOS ONE’s publication criteria as it currently stands. Therefore, we invite you to submit a revised version of the manuscript that addresses the points raised during the review process.

We look forward to receiving your revised manuscript.

Kind regards,

Hikmat Ullah Khan, PhD (Computer Science)

Academic Editor

PLOS ONE

 [This study was supported by a National Research Foundation of Korea (NRF) (http:// nrf.re.kr/eng/index) grant funded by the Korean government (RS-2023-00208278).].  

5. In the online submission form, you indicated that [The datasets utilized or examined in this study can be obtained from the corresponding author upon a reasonable request.]. 

Reviewers' comments:

Reviewer's Responses to Questions

**Comments to the Author**

1. Is the manuscript technically sound, and do the data support the conclusions?

Reviewer #1: Yes

Reviewer #2: Yes

Reviewer #3: Partly

2. Has the statistical analysis been performed appropriately and rigorously? 

Reviewer #1: Yes

Reviewer #2: Yes

Reviewer #3: No

3. Have the authors made all data underlying the findings in their manuscript fully available?

Reviewer #1: Yes

Reviewer #2: Yes

Reviewer #3: No

4. Is the manuscript presented in an intelligible fashion and written in standard English?

Reviewer #1: Yes

Reviewer #2: Yes

Reviewer #3: Yes

5. Review Comments to the Author

Reviewer #1: The research article offers an extensive investigation into topic modeling techniques customized for large language models (LLMs), employing data sourced from LexisNexis and Web of Science. The authors are requested to address the following queries:

1) The manuscript is effectively presented with a strong technical foundation. The methodology description and result discussion sections are compelling and appropriate.

2) The authors should address grammatical errors scattered throughout the manuscript, as well as improve the alignment of equations and enhance the quality of figures to uphold the standards and integrity of the journal.

3) It is advisable for the authors to emphasize the complexity of the proposed method to assess both its robustness and efficiency.

4) When using LLMs for topic modeling, it's essential to consider the computational resources required for processing large volumes of text data and the interpretability of the topics extracted. Additionally, fine-tuning LLMs on domain-specific text corpora may improve the quality of topic modeling results for specialized domains. The authors should address this. Additionally, the authors are suggested to discuss graph-based or deep learning topic modeling techniques to underscore the significance of these approaches.

5) The authors are advised to incorporate state-of-the-art papers in the experimental section for comparative analysis, thus elucidating the significance of their proposed models.

Reviewer #2: > I suggest authors to reconsider the title of this research. If I understood this correctly, this research is about comparing different topic modelling techniques. After reading this particle, I am still having a difficulty in finding on which part of this paper actually deals with "investigating LLM through unsupervised machine learning..."

> The motivation and novelty of this research remain somewhat unclear in the manuscript. The authors state that the goal of this research is to enhance the recognition and derivation of insights from large language models (LLMs) across three distinct platforms. But this research goal is too broad and general. Instead, authors need to specify their aim of research and provide full description on why this research is needed. In addition, it is essential that the Introduction section specifies and clarifies which three unique platforms are under investigation.

> In the Related Works section, the authors present a series of studies on the application of large language models (LLMs) across various domains, and another series of studies on Topic Modeling. This former content might be more appropriate for the Introduction.

> Data Collection section needs to be rewritten with better description and explanation. In the manuscript, authors introduced list of search words, but this is not sufficient. How did authors collect data? Did they use search query for all title, abstract, keyword, etc.? What are the restrictions of data search? any geographical restriction? what

+ Which LexisNexis data source did authors use? Since LexisNexis is well known with its patent database, authors need to clarify this.

> One of my concerns regarding this research is the comparison of topic modelling techniques. LDA and CTM are widely known frequency-based topic modelling technique and their pros and cons are widely known. Unlike them, BERTopic is a topic modelling technique relies on text embedding vector with clustering and data dimension techniques. As the result of this research shows, it is somewhat obvious that BERTopic outperforms compared to other two. In addition, the way BERTopic decides number of topic is totally different from how other two does. Due to this fact, I would suggest authors to compare either frequency-based approaches (such as LDA, DTM, CTM, etc.) or embedding-based approaches.

> Authors need to make a significant improvement in conclusion. Rather than just summarizing the research, authors need to mention how this research contribute to the main stream and its unique value.

Reviewer #3: The paper has a little bit misleading title. I expected from the title, that there will be an overview of LLM technologies themselves, and not an overview of discussion areas of LLM as such. Anyway, the paper may be of interest for a number of readers. I would still recommend extending the title by adding a word: Investigating LLM -> Investigating Discussion Topics Around LLM … or something like this.

1. Is the manuscript technically sound, and do the data support the conclusions?

A drawback of the manuscript is that it is not self-contained. The evaluation metric “diversity” is not defined in the paper. But this metric is crucial when claiming BERT superiority.

The choice of the number of clusters/topics is not explained. I guess that it is based on coherence, but BERT does not have the best coherence at 7 clusters for both datasets. Then why not to choose different number of clusters? In particular, what does it mean that “The crucial point to note is that anomaly occurred when the number of topics was set to three in LexisNexis data.”

Authors use the phrase “Topic 4 was named” etc. but do not tell how the name was given. Was it after invention of authors or was it taken from a predefined list? In the latter case, based on what criterion? The authors used unigrams to describe themes/topics/clusters. Why did they not use bi-grams, tri-grams which could be more informative and possibly better justify the names given to clusters?

Figs 6 and 5 present distances between the topics. But I do not find how these distances were measured and why they fit an Euclidian 2-Dimensional plane? Or did the authors perform a linear/non-linear projection and if so, based on what criteria? I ask this as there are no scales in the figures. The similarity measurers used to create figures 4 and 7 are also unexplained. So we need a definition of distances, of similarities, and how they apply to clusters (are they measured between the cluster center and objects, as average, or 75% point or anything else).

2. Has the statistical analysis been performed appropriately and rigorously?

I do not see any statistical analysis in the paper, but it would be interesting to know if there are statistically significant differences between within and between cluster similarities of documents.

Basic statistics for data should be provided, including: cluster cardinality, mean / median similarity and standard deviation of similarity for each entire set and for the individual clusters. Same would be welcome for diversity and coherence.

*3. Have the authors made all data underlying the findings in their manuscript fully available?

The authors declare that: No - some restrictions will apply.

6. PLOS authors have the option to publish the peer review history of their article (what does this mean?). If published, this will include your full peer review and any attached files.

Reviewer #1: No

Reviewer #2: No

Reviewer #3: No

---

## [Author Response · Author response to Decision Letter 0]

11 Apr 2024

Please refer to re-submitted manuscript for details.

Journal requirements # 1

Thank you for providing guidelines. Based on the PLOS ONE template style you provided, we have reformatted the manuscript to ensure it aligns with PLOS ONE's style requirements. We have also adjusted the file naming accordingly

Journal requirements # 2

Thank you for your valuable guidance. We have publicly shared the code and data used in the experiments on GitHub, following the code sharing policy of PLOS ONE. We intend to describe this in the data availability section. 

Journal requirements # 3

Thank you for providing guidelines. Based on the guidance, we have included the amended role of funder statement in the cover letter regarding the funder's role

Journal requirements # 4

Thank you for your valuable guidance. We have publicly shared the code and data used in the experiments on GitHub, following the code sharing policy of PLOS ONE. We intend to describe this in the data availability section.

Journal requirements # 5

Thank you for your valuable guidance. We have publicly shared the code and data used in the experiments on GitHub, following the code sharing policy of PLOS ONE. We intend to describe this in the data availability section. 

Review #1 Concern #1

Thank you for your valuable comments. Following your suggestions, we have worked on reducing grammatical errors, modifying the numbering and format of the equations and to fit in the PLOS ONE manuscript guidelines. Additionally, efforts were made to enhance the visibility of the figures and the detailed explanations. Only the main points have been included in the letter. Please refer to the manuscript for detailed information

Review #1 Concern #2

Thank you for your valuable comments. Following your suggestion, paragraphs describing each baseline model and BERTopic were added to illustrate the complexity of the proposed methodology and ensure the validity and the efficiency of the approach. Additionally, please note that Fig 2 comparing the models has been added in the Experiment section.

Review #1 Concern #3

Thank you for your valuable statements. Our research focuses not on utilizing LLMs for topic modeling, but on conducting topic modeling for the query "LLM." Nonetheless, we have included discussions on time complexity and the advantages of utilizing BERTopic. To consider the computational resources of the models, we added the time complexity for each model in the "Topic modeling models" section. Comparing the described time complexities, models involving Gibbs sampling such as LDA and CTM require significant resources, while NMF based on matrix factorization and BERTopic, which depends on the number of input data, perform relatively fewer operations. Additionally, NMF, which approximates a given matrix into two low-rank matrices using the document-word matrix, faces difficulty in explicitly representing topics, resulting in low interpretability. On the other hand, the BERTopic model, which considers context during learning, offers high interpretability and clear clustering of topics. Therefore, it is justified for us to use BERTopic to analyze large-scale domain-specific text data.

Review #1 Concern #4

Thank you for your valuable statements. Our research is oriented towards conducting topic modeling for the query "LLM," rather than utilizing LLMs for topic modeling. Nevertheless, we have validated the appropriateness of fine-tuning Modified BERTopic for the text collected using the "LLM" query. Specifically, we implemented a 'modified BERTopic' to analyze data specific to the LLM domain. This is the same as the fine-tuning process. In detail, a revised BERTopic diagram and contents containing these details are attached. This is further explained in the Topic Modeling Models section.

Review #1 Concern #5

Thank you for your valuable statements. This study utilized BERTopic for analysis, and conducted comparisons with two traditionally used models (LDA, NMF) and a deep learning-based approach (CTM) to demonstrate the superiority of the model. The comparative experiments were performed using Coherence and Diversity metrics, where BERTopic performed exceptionally well. Due to its utilization of pre-trained language models like SBERT, BERTopic is categorized within the domain of deep learning methodologies. Furthermore, although it was possible to perform graph-based topic modeling, we did not use that methodology. The reasons are as follows: Firstly, BERTopic generates word and sentence embeddings considering the context, thereby grasping the semantic relationships among words. On the other hand, graph-based learning overlooks the context comprehensively by focusing on relationships between neighboring nodes. Secondly, BERTopic employs clustering algorithms to group similar sentences, whereas graph-based models analyze relationships between nodes, which may not align with the research intent of summarizing topics through clustering. Thirdly, BERTopic extracts important keywords for each topic, enhancing interpretability and facilitating better understanding and summarization of the entire document. For these reasons, we interpreted the data based on BERTopic.

Review #1 Concern #6

Thank you for your valuable comments. We would like to explain the rationale and validity behind our selection of the baseline. We have compared BERTopic with both traditional and relatively new models, and utilized its superiority following the validation process. Specifically, we employed LDA and NMF as baselines due to their wide usage in the field of topic modeling. Additionally, CTM, being one of the newer models and a deep learning-based one, was included in the comparison to verify the suitability of utilizing BERTopic. Secondly, the reason why BERTopic stands out as the optimal choice lies in its flexibility and performance. BERTopic generates context-aware word and sentence embeddings based on the latest deep learning technologies such as SBERT. This allows for a better understanding of semantic similarity between words, and forms topics by clustering similar sentences. These detailed functions have a wide range of applicability because they vary depending on the specific domain, and can be optimized to suit the domain. In fact, other models besides BERTopic have been proposed to date, but BERTopic is the best in terms of applicability. Finally, we added future plans for verification of new SOTA models in the limitations section of the conclusion. If a better and more versatile model than BERTopic is proposed, we will perform additional experiments using those models and compare the results to verify the performance of our model more objectively.

Review #2 Concern #1

We appreciate your valuable comments. Based on your suggestion, we have made an adjustment to the title to better match the content of the research

Review #2 Concern #2

We appreciate your valuable comments. Following your suggestion, we have developed on the research objectives and significance in the last part of the introduction. In detail, the aim of this study is to use machine learning to identify various topics related to LLMs discussed in both media and academia, and to uncover the underlying meanings. The importance of such analysis lies in its value as feedback within industries that utilize LLMs and in its ability to discover new insights. Additionally, it can be utilized when adjusting product or service strategies. 

Review #2 Concern #3

We appreciate your valuable comments. We would like to explain why we believed these aspects are more suitable for the replated works section first, and then we will address the changes that have been made in the manuscript.

The “2.1 Research on utilization of LLMs in various domains” provides insight into the application of LLM across various domains, enabling authors to identify overall trends in LLM and acquire information about ongoing research. Analyzing the “2.2 topic modeling” part is necessary as it is comparable to reviewing the historical aspect of our core methodology in this study. In particular, including the approach of using topic modeling for analysis in the Related Works section confirms the justification for choosing the research methodology and illustrates the benefits of utilizing this method in the study. To provide more detailed information, we have added content to the concluding sections of each related works.

Review #2 Concern #4

We appreciate your valuable comments. Following your advice, we have added additional explanations to the data collection section. Specifically, we have clarified the data collection methods, the range of queries applied, the format of collected data, the necessary conditions for collection, and the platforms used for collection.

Review #2 Concern #5

We appreciate your valuable comments. Firstly, we would like to explain why we used LDA and CTM as baselines, and then inform you that we conducted comparative experiments by adding NMF.

Firstly, LDA is the most widely used topic modeling technique and the most traditional method. This is considered a three-level hierarchical Bayesian model. NMF is a linear algebraic algorithm that uses matrix factorization. Using the term-document matrix as input, the topic is extracted by decomposing it into a term-topic matrix and a topic-document matrix. LDA and NMF modeling techniques perform dimensionality reduction to process complex text, and the BERTopic method we used in this study also includes a dimensionality reduction method called UMAP [14]. Therefore, for the superiority of using BERTopic, it is essential to compare it with similar but traditional techniques such as LDA and NMF. Furthermore, CTM refers to a combined topic modeling approach that combines bag of words with sentence BERT (SBERT). Since CTM also utilizes BERT-based embeddings, it can be considered an embedding-based approach. Therefore, we have included CTM as part of the baseline. Furthermore, since coherence and diversity are metrics used to evaluate the performance of topic modeling regardless of the method employed, we believe it is a valid approach to compare different methods using these metrics [14]. 

Review #2 Concern #6

We appreciate your valuable comments. Based on your advice, we have emphasized the contributions and value of this study in the conclusion section. Specifically, we have described the characteristics of both media and academia, highlighting their differences. Additionally, we have outlined the strengths of utilizing both datasets. Subsequently, we have elaborated on the value that this research can provide and emphasized the value. We have also added references to support these points. 

Review #3 Concern #1

Following your suggestion, we have revised the title to better align with the content of the research.

Review #3 Concern #2

We appreciate your valuable comments. According to the comments you provided, we have added detailed definition, explanations, equation, and information regarding topic diversity, as well as references to support it.

Review #3 Concern #3

We appreciate your valuable comments. Having high coherence indicates that there is a high consistency of words within documents belonging to a single topic. However, when three topics are chosen in topic modeling, it tends to favor larger clusters, limiting the ability to derive a diverse range of topics and resulting in abstract, large-sized clusters. Consequently, conducting detailed analysis on the collected text data becomes impractical. The term "anomaly" was originally used to describe these aspects. However, after reviewing the reviewers' comments, we have added additional model to demonstrate the validity and efficiency of our model selection, and subsequently conducted the additional experiment. Furthermore, to extract better results while reviewing existing literature, we replaced the previously used evaluation metric, c_uci, with c_v and conducted the assessment again, accordingly extracting coherence values. We have also added references supporting these changes. 

Review #3 Concern #4

We appreciate your valuable comments. Based on your concern, we have added explanations and supporting references regarding topic naming. The main theme of each topic was labeled by the authors based on the top keywords with the highest importance included in the topic and from the original data. This method is predominantly used in papers utilizing topic modeling.

Review #3 Concern #5

We appreciate your valuable comments. We have considered using bigrams or trigrams. However, according to previous studies, considering bigrams or trigrams is advantageous for capturing local context since they involve combinations of words (Khoo et al., 2003). On the other hand, unigrams are the most independent units of meaning in sentences, making them more suitable for understanding the overall context. Therefore, we opted for unigrams. Additionally, Tf-idf is a concept that utilizes the inverse proportion of a word's frequency in a specific document and the proportion of documents in which the word appears, and c-tf-idf used in BERTopic extends this concept to focus on extracting class features (Ramos, 2003). Ultimately, since we are considering individual words, unigrams are more appropriate. Below, we have added references for further information.

Review #3 Concern #6

We appreciate your valuable comments. Before addressing your valuable comments, we would like to inform you that we have reordered Figures. Now, let us provide additional explanations for these figures. Firstly, UMAP projects high-dimensional data into two dimensions by finding nearest neighbors and preserving local structure. By reducing to 2D using UMAP, the relative positions of topics can be more clearly expressed, considering the cosine distance between topic embeddings. Subsequently, the number of documents belonging to each topic is represented by the size of circles, which are placed according to the distance between topics. The size of circles indicates the relative size of each topic, reflecting its importance. We have also added this information and reference to the manuscript for better clarity. 

Review #3 Concern #7

We appreciate your valuable comments. We appreciate your valuable comments. Prior to addressing your valuable comments, we wish to notify you that we have rearranged the figures. Now, we would like to offer additional explanations for these figures. The heatmap illustrates the similarity between two topics by representing topic embeddings as vectors and calculating the cosine similarity between each vector pair. Because it involves cosine similarity computation between embedding vectors, the operation is angle-based. We have provided additional clarification and reference on this matter. 

Review #3 Concern #8

We appreciate your valuable comments. Taking your concern into consideration, we have added statistical analysis of evaluation metrics (diversity, coherence values) and cluster cardinality. Additionally, we have provided statistical analysis of topic similarity.

Review #3 Concern #9

We appreciate your valuable comments. We have uploaded minimal anonymized data to a public repository. Furthermore, we will detail this in the data availability section.

---

## [Decision Letter · Decision Letter 1]

16 May 2024

Expansive Data, Extensive Model: Investigating discussion topics around LLM through Unsupervised Machine Learning in Academic Papers and News

PONE-D-24-00612R1

Dear Dr. Kim,

We’re pleased to inform you that your manuscript has been judged scientifically suitable for publication and will be formally accepted for publication once it meets all outstanding technical requirements.

Kind regards,

Prof. Dr. Hikmat Ullah Khan

Academic Editor

PLOS ONE

Additional Editor Comments (optional):

Reviewers' comments:

Reviewer's Responses to Questions

**Comments to the Author**

1. If the authors have adequately addressed your comments raised in a previous round of review and you feel that this manuscript is now acceptable for publication, you may indicate that here to bypass the “Comments to the Author” section, enter your conflict of interest statement in the “Confidential to Editor” section, and submit your "Accept" recommendation.

Reviewer #2: All comments have been addressed

Reviewer #4: (No Response)

2. Is the manuscript technically sound, and do the data support the conclusions?

Reviewer #2: Yes

Reviewer #4: Yes

3. Has the statistical analysis been performed appropriately and rigorously? 

Reviewer #2: N/A

Reviewer #4: Yes

4. Have the authors made all data underlying the findings in their manuscript fully available?

Reviewer #2: No

Reviewer #4: Yes

5. Is the manuscript presented in an intelligible fashion and written in standard English?

Reviewer #2: Yes

Reviewer #4: Yes

6. Review Comments to the Author

Reviewer #2: Dear authors,

I am fully satisfied with all the revised contents as they all are correctly addressed. Thank you for your efforts.

Reviewer #4: The authors have carefully made revisions.

Therefore, the paper is accepted in current form for publication.

7. PLOS authors have the option to publish the peer review history of their article (what does this mean?). If published, this will include your full peer review and any attached files.

Reviewer #2: No

Reviewer #4: No

---

## [Editor Report · Acceptance letter]

20 May 2024

PONE-D-24-00612R1 

PLOS ONE

Dear Dr. Kim, 

I'm pleased to inform you that your manuscript has been deemed suitable for publication in PLOS ONE. Congratulations! Your manuscript is now being handed over to our production team.

Kind regards, 

on behalf of

Dr. Hikmat Ullah Khan 

Academic Editor

PLOS ONE